# Pine Chip and Poultry Litter Derived Biochars Affect C and N Dynamics in Two Georgia, USA, Ultisols

Sharon L. Weyers [1,*], Keshav C. Das [2], Julia W. Gaskin [2] and Amanda M. Liesch [3]

1   North Central Soil Conservation Research Lab, USDA ARS, Iowa Ave., Morris, MN 56267, USA
2   Agricultural and Biological Engineering, University of Georgia, Athens, GA 30602, USA
3   Department of Environmental Engineering Sciences, University of Florida, Gainesville, FL 32611, USA
*   Correspondence: sharon.weyers@usda.gov; Tel.: +1-320-585-8446

**Abstract:** Some biochars produced by pyrolysis of biomass have the potential to sequester C and enhance nutrient supplies in agricultural soils. A 28-day lab incubation was used to assess the potential effects of biochars derived from pine chips (PC) or poultry litter (PL) applied at five application rates (0, 22.5, 45.0, 67.5, and 90 Mg ha$^{-1}$ equivalent). Biochars were applied to two acidic Ultisols, a Cecil sandy loam and a Tifton loamy sand, found in Georgia, USA. Cumulative basal soil respiration was measured over the 28-day incubation. Other soil properties measured before and after incubation were soil pH, total soil organic carbon (SOC), total soil N, soluble organic C (OC), soil mineral nitrogen (NH$_4^+$-N and NO$_3^-$-N), and microbial biomass C (MBC). Before incubation, addition of both PC and PL biochars increased soil pH, total SOC, and C:N ratio in both soils. Addition of the PL biochar increased total soil N, soluble OC, and NO$_3^-$-N in both soils, MBC in Tifton soil, and NH$_4^+$-N in Cecil soil. Addition of the PC biochar decreased NO$_3^-$-N in Cecil soil but increased it in Tifton soil. After the 28-day incubation, averaged across soils, pH increased in the 22.5 Mg ha$^{-1}$ PC and 22.5 and 67.5 Mg ha$^{-1}$ PL treatments, total SOC declined in the 45 and 67.5 Mg ha$^{-1}$ PC treatments, and the C:N increased in soil controls and decreased in the 67.5 Mg ha$^{-1}$ PC treatment. In Cecil soil, the MBC declined in PL treatments except at 90 Mg ha$^{-1}$, and NH$_4^+$-N declined in the 90 Mg ha$^{-1}$ PC treatments. In Tifton soil, MBC increased in the 45 Mg ha$^{-1}$ PL treatment, and NH$_4^+$-N increased in all but the 22.5 Mg ha$^{-1}$ PL treatments. Total N and NO$_3^-$-N did not change with incubation. Basal respiration was not affected by biochar, thought it was generally greater in Cecil than Tifton soil. Net SOC loss and the initial increase in soluble OC and MBC indicated potential C priming from adding both biochars. Increased NH$_4^+$-N with time in Tifton PL treatments indicated potential N priming. In Cecil soil, the PC biochar may have immobilized NH$_4^+$-N, but PL biochar likely supplied it. In Tifton soil, PC biochar appeared to be generally inert, but PL biochar supplied soluble OC and NH$_4^+$-N, although it might have inhibited nitrification.

**Keywords:** pyrolysis; soil amendments; carbon sequestration; mineralization; priming effect

## 1. Introduction

Biochar is often promoted as a soil amendment that can improve soil quality and soil carbon sequestration [1,2]. This usage followed the discovery that long-term addition of organic matter as charcoal (i.e., black carbon or biochar) and other refuse improved the soil quality of Terra Preta soils in the Amazon basin [2,3]. Biochar is resistant to degradation as it has been found in ancient sedimentary rocks [4], and is estimated to have a soil residence time of 250 to 3280 years [5]. Specific use of biochar in agricultural systems is prescribed for the purpose of increasing crop yields, improving soil health, and mitigating environmental externalities of intensive production, such as climate change, or soil C loss by increasing soil C [6]. Alternatively, biochar can be used in various environments to remove contaminants [7,8].

Improved soil health can be measured as an increase in soil organic matter, biological activity, or availability of nutrients for crop uptake, among other aspects [9]. Recent studies showed increased microbial activity within months to years following addition of manufactured biochar or wildfire-derived charcoals [10–13]. These impacts indicate that biochar functions like crop or animal residues, which typically increase microbial biomass and nutrient cycling [14]. Microbial activation implies that biochar provides a nutritious substrate for microbes [15]. However, microbial activation might also be due to increases in water holding capacity, provision of surfaces for microbial attachment or refuge, or other physio-chemical changes [16–18]. Studies indicated that microbial stimulation is due to a C priming effect, whereby the addition of biochar increased the microbial turnover, measured as mineralization of C into $CO_2$, of either the added biochar or native soil organic matter [11,19–21]. On the other hand, N priming effects might also be possible and would be indicated by N turnover, measured as increased ammonification or nitrification processes [22]; N priming would have positive outcomes for soil fertility and crop production.

In agricultural systems, one of the desired uses of biochar amendments is to improve crop production. Poultry litter biochar has been promoted for this purpose due to its high nutrient content [23]. Benefits might be due to biochar-induced increases in cation exchange capacity (CEC) or pH, which improve soil nutrient cycling processes [24]. On the other hand, many biochars contain ash, which can be a direct source of nutrients [25]. Ash content is influenced by the source biomass used for pyrolysis, by recipient soil properties, and by the weather when applied [1]. Two recent meta-analyses indicated that yield and associated soil health benefits from biochar are more likely to occur with application to degraded soils with a low cation exchange capacity, low organic C content, and low pH, such as highly weathered tropical soils rather than fertile temperate soils [26,27]. On the other hand, these meta-analyses also revealed that yields increased on average by only 10% and did not respond to increases in application rate; further yield losses might also occur. Considering these results, the use of small-scale laboratory incubations might assist in determining appropriate biochar selection and field application rates to improve the cost effectiveness of deployment.

In the present study, we measured changes in multiple processes in two different Georgia Ultisols after adding increasing amounts of two contrasting biochars. The two primary goals were (1) to determine if responses in C and N dynamics depended on soil type, biochar type, or application rate; and (2) to assess potential biochar priming effects that might affect suitability for field application through delivery of nutrients or carbon sequestration properties. A secondary goal was to evaluate the utility of the short-term incubation to provide important information on impacts, which will inform field deployment for further research or prescribed use.

## 2. Materials and Methods

### 2.1. Experimental Set Up

Mesocosm incubations were used to assess biochar effects on treated soil substrates. The substrate soils used were two Ultisols that varied in pH, texture, and organic matter content. The soils were a fine textured sandy loam Cecil soil (fine, kaolinitic, thermic Typic Kanhapludult) and a coarser textured loamy sand Tifton soil (fine-loamy, kaolinitic, thermic Plinthic Kandiudult). The soils selected are common in the Piedmont and Coastal Plain geographic regions in the state. Cecil and Tifton soils were collected at University of Georgia Experiment stations in Watkinsville, GA, and Tifton, GA, respectively. The soils were excavated from 0 to 15 cm depth, sieved to remove coarse debris, and then air dried. Approximately $375 \pm 0.1$ g of the Cecil soil or $450 \pm 0.1$ g of the Tifton soil were placed in individual cylindrical incubation chambers with an enclosed bottom to prevent leaching, each 18 cm in height and 6.4 cm in diameter. Resulting soil depths were ~12–13 cm and had surface areas of 32.2 cm². A total of 144 chambers, 72 per each soil type, were created. Eight chambers for each soil were reserved as non-amended soil type controls.

The biochars were selected due to the potential availability of two feedstocks from prominent paper and poultry production industries in the state. The biochars were obtained by pyrolyzing pine chips (PC) or poultry litter (PL) feedstocks at a maximum temperature of 400 °C, a holding time of 0.5 h, and $N_2$ as a carrier gas. They were produced in a similar process as described and evaluated in Gaskin et al. [28]. The PC biochar had a 73% C, 0.15% N, 1.4% ash, and 26% volatile content; a pH of 7.32; an electrical conductivity (EC) of 0.112 mS; and a C:N ratio of 490. The PL biochar had a 42% C, 4.24% N, 53% ash, and 22% volatile content; a pH of 10.27; an EC of 15.5 mS, and a C:N of 10. Additional macro- and micro-nutrient properties of the two biochars were reported by Liesch et al. [29]. The cation exchange capacity was estimated from the reported content of calcium, magnesium, sodium, and potassium in milliequivalents per 100 g of biochar, and was 35 for PC, and 747 for PL. Approximately 95% of each material was 1–2 mm in size. Both PC and PL biochars were applied at 7, 14, 21, and 28 g to each of eight replicate chambers for each of the two soils in the incubation chambers. These application weights equated to 22.5, 45, 67.5, and 90 Mg ha$^{-1}$, based on the surface area and a 15 cm depth. Thus, the biochar:soil ratios ranged from 1.9 to 7.4% for the Cecil soil treatments, and 1.6 to 6.3% for the Tifton soil treatments. An application rate of 45 Mg ha$^{-1}$ was set as the 100% application rate reference point, as this was the most reported field application rate in the literature. Therefore, the nine final treatment rate designations, including the no-char (NC) controls, were: NC0, PC50, PC100, PC150, PC200, PL50, PL100, PL150, and PL200.

For the duration of the experiment, incubation chambers were kept covered with parafilm at a constant temperature of 22 °C, in the dark, and maintained at 26% moisture by periodic weight measurements and adding deionized water if necessary. After an initial five-day pre-incubation, half of the 144 chambers, representing four replicates for each of the nine treatments and both soils, were deconstructed to obtain an initial baseline. The remaining chambers were assayed for final properties after an additional 28 days of incubation.

*2.2. Sampling and Analyses*

At each of the initial and final sampling times, all chambers were deconstructed, soils were sieved and subsampled for wet chemistry and soil moisture measurement (105 °C); the remainder was air-dried and ground with a mortar and pestle for dry analyses.

Microbial biomass-associated C (MBC) was determined from the additional flush of C induced by chloroform fumigation of paired moist soil samples extracted with 0.5 M potassium sulfate ($K_2SO_4$) and adjusted for extraction efficiency with a $k_{ec}$ = 0.38 [30]. Briefly, the paired subsamples of approximately 15 g dry weight of soil were taken, one sample was immediately extracted with 20 mL of 0.5 M $K_2SO_4$ the other sample underwent chloroform fumigation over 4 days before similar extraction. Extracts of both fumigated and unfumigated samples were analyzed for soluble organic C (OC) and mineral N forms. Soluble total C and inorganic C in extracts were measured by combustion with a Lachat IL550 TOC-TN. Soluble OC was calculated as the difference between total and inorganic forms measured in the same extraction solution.

Soil total C and total N were measured by dry combustion with a Leco CN 2000 analyzer. Soil organic C (SOC) was determined by subtraction of the inorganic C (SIC) component from total C measured by pressure formation after the addition of HCl [31]. Soil pH was measured in a 1:2 soil to water mixture containing $CaCl_2$.

Soil solubilized inorganic N forms, nitrate plus nitrite ($NO_3^- + NO_2^-$) and ammonium ($NH_4^+$), were measured in the 0.5 M $K_2SO_4$ extracts from the non-fumigated soil sub-sample obtained with the microbial biomass analyses described above. These forms were obtained with standard colorimetric methods, as described by Mulvaney [32], using an Alpkem auto-analyzer. Nutrient concentrations are reported in g kg$^{-1}$ soil or mg kg$^{-1}$ soil.

Microbial activity was determined by basal soil respiration measurements taken once every seven days starting on day 0, when baseline measurements were taken, through day 28. A subset of three replicates for each of the nine treatments were monitored

weekly. At each time, the soil chambers were capped for four hours, and headspace gas was sampled through a permanent septum inserted in the cap. Accumulated $CO_2$ in the headspace sample was measured by thermal conductivity detection on a Varian CP-3800 GC-TCD. Respiration per chamber, measured as µg $CO_2$-C hr$^{-1}$, was adjusted for headspace volume. Total basal soil respiration in total g $CO_2$-C per kg$^{-1}$ soil emitted over 28-days was calculated for each replicate chamber by summing the estimated average daily rate for each week assuming a linear change from week to week. This was done by calculating daily respiration at the midpoint of the slope (ca. day 4) from day 0 to day 7 of the week. This is equivalent to calculating the area under the curve generated by the graph of respiration over time.

### 2.3. Statistical Analyses

Impacts on the six soil properties (pH, total SOC, total soil N, C:N ratio, soluble OC, MBC, $NH_4^+$-N, and $NO_3^-$-N) were evaluated. The statistical design used was a completely randomized design with sample time, soil type, and treatment as the main fixed effects and replication as a random effect. The sample time had two levels, initial (assessed at 5-days following preincubation), and final (assessed after an additional 28-day incubation). The soil type had two levels, Cecil and Tifton soils. The treatment effect had nine levels: NC0, PC50, PC100, PC150, PC200, PL50, PL100, PL150, and PL200. A general linear mixed model (Proc GLIMMIX) was used to evaluate the main effects and all possible two-way and the three-way interactions. If the three-way interaction was not significant ($p > 0.05$) the model was reduced to main effects and two-way interactions and reanalyzed. Multiple comparisons were made within significant two-way or three-way interactions as necessary, and the *slicediff* function was used to focus only on main effects of time, soil, and treatment factors. A log normal distribution was necessary to adjust for non-normality for most soil properties except pH, $NH_4^+$-N, and $NO_3^-$-N, which were normally distributed. A similar approach was used to assess basal soil respiration; however, since total accumulated basal soil respiration was analyzed, the only main effects were treatment and soil type. All main effects, interactions, and multiple comparisons were considered significant at $p < 0.05$. All statistical analyses were conducted with SAS release 9.4 [33].

### 3. Results

### 3.1. Overview of Significant Effects

Soil properties did not change over time for most of the measured soil properties (Table 1). Change over time for total soil OC and $NO_3^-$-N was influenced by significant interactions with treatment, but for total soil, N was independent of any other effect. As expected, soil type and treatment conditions (i.e., type of biochar and application rate) had a significant impact on soil properties, but in most cases, this impact was influenced by significant two-way or three-way interactions (Table 1).

**Table 1.** Significant parameters of main and interaction effects for soil properties and basal soil respiration.

| Effect | pH$_{CaCl}$ | Total SOC | Total Soil N | C:N | solOC | MBC | $NH_4^+$-N | $NO_3^-$-N | $CO_2$-C |
|---|---|---|---|---|---|---|---|---|---|
| Time | ns | * | * | ns | ns | ns | ns | ns | NA |
| Soil | *** | *** | *** | *** | ns | *** | ns | *** | *** |
| Treatment | *** | *** | *** | *** | *** | *** | *** | *** | ** |
| Time*soil | ns | ns | ns | ** | *** | * | *** | ns | NA |
| Time*treatment | ** | * | ns | ** | ** | ns | *** | ** | NA |
| Soil*treatment | ** | *** | ** | *** | ** | *** | *** | *** | ** |
| Time*soil*treatment | ns | ns | ns | ns | * | *** | *** | ns | NA |

pH$_{CaCl}$—pH measured in calcium chloride; SOC—soil organic C; solOC—soluble organic C; MBC—microbial biomass C; ammonium—$NH_4^+$-N; nitrate—$NO_3^-$-N; total soil respiration—$CO_2$-C. ns—not significant; NA—not applicable; * $p < 0.05$; ** $p < 0.01$, *** $p < 0.001$.

### 3.2. Soil pH

Soil pH was influenced by the main effects of soil type and treatment, and other two-way interactions (Table 1). Prior to adding biochar, the pH averaged 5.64 across Cecil and Tifton soils (Table 2). The application of PC biochar had a moderate effect on pH across soils, elevating it from pH 5.6 in NC0 to a high of pH 5.8 in PC100 and PC150, whereas the application of PL biochar significantly increased pH at increasing application rates up to 8.95 in PL200 (Table 2). However, PC biochar had a slight acidifying effect on pH in Cecil soils at lower application rates, but the opposite effect in Tifton soils (Table 3). A change over time, averaged across soils, occurred only in PC50, PL50, and PL150.

**Table 2.** Means of soil properties showing significant differences among treatments and soils for significant time*treatment interactions.

| Treatment | pH$_{CaCl}$ | | Total SOC g kg$^{-1}$ Soil | | Total N g kg$^{-1}$ Soil | | C:N | |
|---|---|---|---|---|---|---|---|---|
| | Initial | Final | Initial | Final | Initial † | Final † | Initial | Final |
| NC0 | 5.64 hi | 5.56 ij | 6.9 k | 8.3 j | 0.96 | 0.96 | 7.0 j | 8.4 i |
| PC50 | 5.47 j | 5.64 hi | 19.8 gh | 18.2 h | 1.02 | 0.99 | 18.7 e | 18.4 e |
| PC100 | 5.82 g | 5.77 gh | 33.7 d | 29.2 ef | 1.04 | 0.99 | 31.9 d | 29.1 d |
| PC150 | 5.80 g | 5.81 g | 46.7 b | 40.7 c | 1.04 | 1.02 | 43.7 b | 39.4 c |
| PC200 | 5.70 gh | 5.73 gh | 54.6 a | 51.0 a | 1.04 | 1.00 | 52.7 a | 52.1 a |
| PL50 | 7.47 f | 7.64 e | 13.7 i | 12.8 i | 1.38 | 1.28 | 9.7 h | 10.0 gh |
| PL100 | 8.18 d | 8.14 d | 21.2 g | 19.9 gh | 1.85 | 1.84 | 11.3 f | 10.7 fg |
| PL150 | 8.63 b | 8.46 c | 26.1 f | 26.1 f | 2.25 | 2.31 | 11.5 f | 11.3 f |
| PL200 | 8.95 a | 8.85 a | 33.8 d | 30.2 de | 2.84 | 2.67 | 11.8 f | 11.3 f |

Different lowercase letters indicate significant differences among time*treatment combinations within a soil property. A dagger (†) indicates data that is provided for completeness but not valid for multiple comparison due to lack of significant two-way interaction.

**Table 3.** Means of soil properties showing significant differences among treatments and soils for significant soil*treatment interactions.

| Treatment | pH$_{CaCl}$ | | Total SOC g kg$^{-1}$ Soil | | Total N g kg$^{-1}$ Soil | | C:N | |
|---|---|---|---|---|---|---|---|---|
| | Cecil | Tifton | Cecil | Tifton | Cecil | Tifton | Cecil | Tifton |
| NC 0 | 6.20 g | 4.99 l | 10.4 h | 4.9 i | 1.19 f | 0.73 h | 8.7 k | 6.7 l |
| PC50 | 5.93 h | 5.17 k | 25.8 d | 12.3 g | 1.28 f | 0.73 h | 20.2 f | 16.9 g |
| PC100 | 5.94 h | 5.65 i | 41.3 b | 21.5 e | 1.28 f | 0.75 h | 32.4 d | 28.7 e |
| PC150 | 6.00 h | 5.61 i | 59.6 a | 27.8 d | 1.30 f | 0.75 h | 46.3 b | 36.8 c |
| PC200 | 5.99 h | 5.43 j | 66.2 a | 39.4 b | 1.30 f | 0.74 h | 51.0 a | 53.8 a |
| PL50 | 7.61 f | 7.50 f | 16.8 f | 9.7 h | 1.59 e | 1.07 g | 10.6 ij | 9.1 k |
| PL100 | 7.99 e | 8.34 d | 25.5 d | 15.6 f | 2.20 c | 1.49 e | 11.6 hi | 10.4 j |
| PL150 | 8.45 d | 8.63 c | 31.7 c | 20.5 e | 2.72 b | 1.84 d | 11.7 hi | 11.1 hij |
| PL200 | 8.79 b | 9.01 a | 39.2 b | 24.8 d | 3.30 a | 2.20 c | 11.8 h | 11.3 hij |

Different lowercase letters indicate significant differences among soil*treatment combinations within a soil property.

### 3.3. Total SOC and N, and C:N Ratio

Total SOC was significantly affected by all main effects and two-way interactions of treatment with both time and soil (Table 1). Averaged across all treatments, the initial total SOC was 28.5 g kg$^{-1}$ soil compared to 26.3 g kg$^{-1}$ soil after the 28 day incubation; however, the interaction of time*treatment indicated that changes averaged across soil type were significant only where it increased in the NC0 control, and where it decreased in PC100 and PC150 (Table 2). Total SOC averaged over time and treatments was greater in Cecil soil (35.2 g kg$^{-1}$) than Tifton soil (19.6 g kg$^{-1}$), and was a consistent difference between soils within each treatment (Table 3). The significant interaction of soil*treatment was due to similarities across treatments and soils, rather than within a treatment between soils. For

example, Tifton PL200 had a similar amount of SOC as Cecil PC50 and PL100 and Tifton PC150. Additionally, total SOC increased with increasing application rates of each of the biochars within the initial and final sampling times in Cecil and Tifton soils, except for a similarity between initial PC150 and PC200 treatments.

Total soil N was significantly affected by all main effects but there was only one two-way interaction of treatment with soil (Table 1). In the absence of a significant two-way interaction with time, soil N decreased by only 0.04 g kg$^{-1}$ soil from an initial 1.49 g kg$^{-1}$ soil to a final 1.45 g kg$^{-1}$ soil, averaged over treatments and soils. As occurred with SOC, differences between the two soil types were also consistent, whereby total soil N was greater for all treatments in the Cecil soil than in the Tifton soil (Table 3). In part, this was due to background differences in soil N being greater in Cecil NC0 controls than in Tifton NC0 controls. Additionally, the application of PC biochar had no effect on total soil N in either soil, but the application of PL biochar increased total soil N in both soils.

The patterns observed with the C:N ratio were different than total SOC or N in that time was not a significant main effect, but all three two-way interactions were significant (Table 1). The C:N ratio was greater in Cecil than in Tifton soils but changed over time only in Cecil soils from 22 to 24 and remained at 21 in Tifton soils. By averaging the treatment over soils, a change over time occurred only in NC0 and PC150 (Table 2). Additionally, C:N increased with an increasing application rate of PC in both soils but was only moderately affected by PL application in Tifton soil (Table 3).

*3.4. Soluble and Microbial Soil Properties*

Soluble OC was influenced by the main effect of treatment and all two-way and three-way interactions (Table 1). Soluble OC at the initial and final time points within a soil and treatment increased in the Cecil NC0 and Cecil PL50 and decreased in Tifton PL200 (Table 4). Soluble OC between soils within a treatment and time point was greater in Tifton than Cecil soil for the initial PC50 and the final NC0 controls, but greater in Cecil than Tifton soil for PL50. Soluble OC among treatments was consistently greater in all PL treatments than in PC or NC treatments, regardless of time point or soil type, but had no consistent response to increasing application rates (Table 4).

MBC was significantly affected by most main and interaction effects except time and time*treatment (Table 1). In Cecil soil, the initial MBC was unaffected by treatment with either biochar; over time, MBC decreased in PL50 and PL100 and increased in PL150 and PL200; and the final MBC was similar among NC0 and PC treatments and increased with an increasing application rate of PL (Table 4). In Tifton soil, the initial MBC was moderately elevated by application of PC char, and substantially increased at increasing application rates of PL biochar; changes over time were limited to PL100; and the final MBC was similar among NC0 and PC treatments, but no longer increased with the increasing application rate of PL. Comparing soil types, the initial MBC was consistently greater in Cecil over Tifton soils for all but PL200, and the final MBC was greater in Cecil than Tifton soils for all NC and PC treatments, but only in the PL200 among PL treatments.

Impacts on NH$_4^+$-N were dependent on the significant three-way interaction of all main effects (Table 1). In Cecil soil, initial NH$_4^+$-N was greater in PC 200, PL50, and PL100 compared to the NC0 control (Table 4). In Tifton soil, biochar application had no effect on initial NH$_4^+$-N. With the 28-day incubation, NH$_4^+$-N decreased in Cecil PC100, PC150, and PC200 and increased in Tifton PL100, PL150, and PL200. In both soils, the final NH$_4^+$-N was greater in most PL treatments than PC treatments.

In contrast to NH$_4^+$-N, NO$_3^-$-N was not influenced by the three-way interaction but was by the time*treatment and soil*treatment interactions (Table 1). Regarding the soil*treatment interaction, NO$_3^-$-N averaged over initial and final samples within the Cecil soil was greater in NC0 compared to all PC treatments and increased with increasing application rates of PL biochar (Table 4). In Tifton soil, NO$_3^-$-N averaged over sampling points was greater in PC treatments compared to NC0 but also increased with increasing application rates of PL biochar. Regarding the time*treatment, NO$_3^-$-N averaged across

soil types was significantly increased with incubation in PC50 (5.5 to 5.6 mg kg$^{-1}$ soil) and PL50 (7.5 to 7.6 mg kg$^{-1}$ soil) and decreased with incubation in PL150 (8.6 to 8.5 mg kg$^{-1}$ soil); no other changes occurred.

**Table 4.** Means of soluble OC (solOC), microbial biomass C (MBC), $NH_4^+$-N, and $NO_3^-$-N showing simple main effects analyzed within the significant three-way interactions of time*soil and type*treatment. Treatments were no char (NC), pine chip biochar (PC), or poultry litter biochar (PL) at different application rates, as indicated.

| Treatment | solOC | | MBC | | $NH_4^+$-N | | $NO_3^-$-N | | |
|---|---|---|---|---|---|---|---|---|---|
| | Initial | Final | Initial | Final | Initial | Final | Initial † | Final † | Average ‡ |
| | | | | | mg kg$^{-1}$ | | | | |
| | | | | | Cecil | | | | |
| NC0 | 39 *c | 109 *#b | 406 #a | 385 #bc | 6.6 c | 12.5 #b | 6.3 | 6.1 | 6.2 #f |
| PC50 | 38 #c | 52 c | 408 #a | 383 #b | 7.6 bc | 1.6 #c | 5.9 | 6.0 | 5.9 #e |
| PC100 | 44 c | 46 c | 394 #a | 418 #b | 12.2 *bc | 0.7 *c | 5.9 | 6.0 | 5.9 #e |
| PC150 | 45 c | 48 c | 352 #a | 356 #bc | 9.0 *bc | 0.6 *#c | 6.0 | 6.0 | 6.0 #e |
| PC200 | 51 c | 48 c | 339 #a | 368 #bc | 14.7 *ab | 0.7 *#c | 6.0 | 6.0 | 6.0 #e |
| PL50 | 121 *b | 218 *#a | 432 *#a | 213 *d | 20.6 #a | 17.0 #a | 7.5 | 7.7 | 7.6 d |
| PL100 | 188 a | 169 a | 430 *#a | 265 *d | 21.4 *#a | 3.0 *#c | 8.0 | 8.0 | 8.0 #c |
| PL150 | 165 a | 196 a | 388 *#a | 284 *cd | 11.7 #bc | 15.0 ab | 8.6 | 8.3 | 8.4 #b |
| PL200 | 191 a | 178 a | 423 *a | 598 *#a | 11.8 #bc | 16.9 a | 8.9 | 8.7 | 8.8 #a |
| | | | | | Tifton | | | | |
| NC0 | 45 c | 36 #de | 81 #f | 82 #de | 8.4 a | 3.5 #d | 5.0 | 5.0 | 5.0 #h |
| PC50 | 53 #c | 52 cde | 81 #f | 73 #e | 9.5 a | 11.3 #c | 5.1 | 5.3 | 5.2 #g |
| PC100 | 49 c | 43 cde | 101 #ef | 112 #c | 6.8 a | 5.7 cd | 5.7 | 5.6 | 5.6 #f |
| PC150 | 42 c | 54 c | 116 #de | 99 #cd | 5.8 a | 10.1 #cd | 5.6 | 5.6 | 5.6 #f |
| PC200 | 46 c | 35 e | 96 #ef | 87 #cd | 7.5 a | 8.7 #cd | 5.4 | 5.5 | 5.4 #e |
| PL50 | 138 b | 127 #b | 159 #cd | 184 b | 8.1 #a | 9.0 #cd | 7.5 | 7.5 | 7.5 d |
| PL100 | 240 a | 202 a | 185 *#bc | 306 *ab | 7.7 *#a | 29.0 *#a | 8.4 | 8.3 | 8.3 #c |
| PL150 | 149 b | 153 a | 230 #b | 237 b | 2.4 *#a | 16.3 *bc | 8.7 | 8.6 | 8.6 #b |
| PL200 | 263 *a | 168 *a | 371 a | 368 #a | 3.3 *#a | 23.1 *ab | 9.0 | 9.0 | 9.0 #a |

Asterisk (*) indicates a significant difference between initial and final values of pairs with the same soil type and treatment (same row). Hashtag (#) indicates a significant difference between soil types of pairs with the same time point and treatment (same column different soil). Different lowercase letters indicate a significant difference among treatments within the same time point and soil type (same column same soil). Dagger (†) indicates data that is provided for completeness but not valid for statistical analysis due to lack of significant three-way interaction. Double dagger (‡) indicates the multiple comparison outcome of the two-way soil*treatment interaction.

### 3.5. Basal Soil Respiration

As a measure of microbial activity, cumulative basal soil $CO_2$-C respired over the 28-day incubation in controls and all application rates of biochar was influenced by the main effects of soil and treatment and their interaction (Table 1). Multiple comparisons within the soil*treatment interaction indicated that differences in total cumulative soil respiration were primarily due to soil type, whereby respiration within a treatment was greater in Cecil than Tifton soil, except for PC200 and PL50 (Table 5). In Cecil soil, respiration was greater in PL100 and PL150 than PC150 and PC200 and was not influenced by increasing the application rate. In Tifton soil, differences among application rates did occur, for example, respiration was significantly lower in PC150 than in the five other treatments, but differences were not directional with the application rate.

**Table 5.** Total soil respiration (total g CO2-C kg soil) emitted over a 28-day incubation.

| Treatment | Soil | |
|:---:|:---:|:---:|
| | Cecil | Tifton |
| NC0 | 1.10 abcd | 0.38 fghe |
| PC50 | 1.04 abcd | 0.53 efgh |
| PC100 | 1.19 abc | 0.25 hij |
| PC150 | 0.71 bcde | 0.15 j |
| PC200 | 0.63 cdef | 0.44 efg |
| PL50 | 1.43 abc | 0.69 cde |
| PL100 | 1.42 a | 0.18 hij |
| PL150 | 1.50 a | 0.64 def |
| PL200 | 1.39 ab | 0.24 ghij |

Lower case letters that differ indicate differences among treatments and soils ($p < 0.05$).

## 4. Discussion

### 4.1. Carbon Dynamics

The carbon dynamics evaluated in the current study differed depending on soil and biochar. This was expected, as different types and quantities of biochar applied to a variety of soils have induced various changes in soil C dynamics [34]. Across soils, both biochars added C, but total SOC was lost at all application rates with PC, but only at the highest application rate with PL. In Cecil soils, PL biochar increased soluble OC, but both biochars seemed to inhibit further changes. The initial MBC was not changed with either biochar, but it did decrease after incubation at most PL application rates. In Tifton soil, PC biochar had no effect on soluble OC dynamics, and a minor effect on MBC, but PL biochar increased soluble OC and MBC at additional rates or over time. Potential mechanisms for these differences included soil pH changes, biochar-soil sorption interactions, and nutrient supply by the biochar.

A loss in total SOC after the 28-day incubation was observed in two treatments receiving PC biochar. This indicated that this biochar might not be always useful for the purpose of carbon sequestration suggested by others [1,2], and that there is a need to be selective with the type of biochar used as an amendment for a particular soil. In contrast, soil SOC was not lost from PL biochar applications. The higher degree of degradation in Tifton soil, suggested by a lower initial SOC, supports the findings of Crane-Droesch et al. [26] and Jeffery et al. [27] that soil C is more likely to be increased and sequestered if biochar is applied to a degraded rather than a fertile soil.

Amending soil with biochar was expected to influence initial content and changes over time in soil soluble OC. These expectations are based on findings that the addition of fresh and aged wood- and grass-derived biochars, produced at 250–650 °C, increased water dissolved OC [35]. Although the PC biochar was a wood-derived biochar made at 400 °C, it did not influence the initial soluble OC. On the other hand, PL biochar did increase the initial soluble OC, probably because the poultry litter feedstock also contains wood-based bedding materials. Previous biochar studies have found that soluble OC pools increase with time (e.g., [19,35–37]). This is likely because soluble OC is a labile C pool that is both an indicator and regulator of microbial C processes [38–40]. In contrast, the only increases in soluble OC with incubation were observed in the Cecil NC0 and PL50 treatments. The lack of change in soluble OC might indicate inhibition rather than priming of microbial processes. However, the biochars or soils used might also have absorbed any processed soluble OC. Chemical binding or sorption of soluble OC is possible and more likely in finer-textured soils where organic–mineral bridging can occur [41–44].

The initial increase in MBC with PL addition to Tifton soil, but limited positive or negative changes following incubation, partly agrees with the findings of similar studies. A literature review revealed that the addition of animal manure and wood derived biochars to a range of different soils stimulated soil microbial communities in many short-to-long-term lab experiments [45]. Further, the addition of a bovine manure/pine shavings derived biochar resulted in an increased microbial biomass and basal soil respiration at increasing

application rates when applied to similarly textured sandy loams and loamy sand soils [12]. More specifically, the addition of a poultry litter biochar at 5 Mg ha$^{-1}$ to a loamy sand significantly increased the microbial biomass and basal respiration [46]. These findings contrast with the observed loss in MBC that occurred in the Cecil soil with PL addition at some application rates. However, this loss might be an artifact of the fumigation procedure as it may underestimate MBC [47,48]. This would be the more probable scenario as the average loss of 163 mg MBC kg$^{-1}$ soil in the Cecil PL did not equate to a measurable change in SOC or amount of respired total g $CO_2$-C kg$^{-1}$ soil.

The absence of an increase in the basal soil respiration with biochar application contradicts the results of a similar investigation where, cumulatively, it was elevated after 28 days by 6% or 59%, respectively, following pine chip and poultry litter derived biochars produced at 350 °C and applied at 2 g to 200 g of a similar loamy sand Norfolk Ultisol, classified as a fine-loamy, kaolinitic, thermic Typic Kandiudult [49]. The current findings do not reflect the potential for increased soil respiration observed up to 50 days with increasing application rates up to 44.8 Mg ha$^{-1}$ of a switchgrass-derived biochar produced at 500 °C to a silty loam soil with an initial C content of 0.7% and pH of 5.4 [13]. However, the current findings for PC biochar were somewhat in line with the decrease in soil respiration observed at 6 and 18 months following increasing applications up to 30 t ha$^{-1}$ of corn stover biochar produced at 600 °C to a silt loam soil [50,51]. No related reports were found for decreased or inhibited respiration at increasing application rates of biochars to similar soils and measured within a month after application.

*4.2. Nitrogen Dynamics*

Low N content is typical of wood-based feedstocks that have very high initial C:N ratios, whereas manure-based feedstocks have lower C:N due to nitrogenous waste content, particularly, uric acid or urea, although some contributes to ammonia volatilization in low temperature pyrolysis biochars (≤400 °C) [52]. Not surprisingly, PL application induced more of the initial changes in total soil N than PC biochar. Despite the lack of treatment effects, the overall loss of 0.04 g N kg$^{-1}$ soil suggested greater loses are possible with the addition of either PC or PL biochar on a larger scale. As leaching of mineral N was prevented by the closed chamber incubations, the logical avenue of total N loss could be N gas emissions. As suggested by other findings, gaseous loss of N as nitrous oxide ($N_2O$) through microbial processes, if it occurred, might not have been enough to account for the total N loss observed here. For example, observations of net $N_2O$-N emissions were estimated to be less than 2.0 mg kg$^{-1}$ soil over a 28-day incubation, following application of up to 50 Mg ha$^{-1}$ of biochars derived from green waste, poultry litter, papermill waste, or biosolids to an Australian Ferrosol [53]. The more probable route of N loss would be ammonia ($NH_3$) emissions, which is already a particular concern for alkaline soils [54], but has also been found when applying high pH biochars to acidic soil [55]. One study determined that when soil pH was below 8.2, poultry manure-based biochar, compared to green waste or wheat straw biochars, reduced $NH_3$ emissions by up to 52% [56]. This instance would favor reduced emissions in any current treatment other than PL150 or PL200 where the pH was above 8.4. Surprisingly, the approximated 30-day cumulative admission rates were less than 150 mg $NH_3$ kg$^{-1}$ soil with 0.4 g poultry manure biochar applied to 20 g soil, but 210 mg $NH_3$ kg$^{-1}$ soil when applied at half that application rate [56]. These values would approximate to a loss of 124 and 172 mg N kg$^{-1}$ soil, respectively, over 30 days.

Nitrogen dynamics differed between Cecil and Tifton soils. However, various outcomes of increased, decreased, or no change in N mineralization or nitrification processes can be expected with the application of different biochars and soil conditions [34]. The initial increase in $NH_4^+$-N in Cecil soils from PL biochar, and elevation with PC biochar, was potentially due to mineralization of native soil organic matter, as neither biochar added significant amounts of $NH_4^+$-N in Tifton soil at the initial application. With incubation, ammonification declined in Cecil soils but was elevated or increased in Tifton soils with

incubation. In contrast, $NO_3^-$_N availability was inhibited in Cecil soils with PC biochar but increased with PL biochar, and in Tifton soil, it increased with the application rate of both biochars. Despite the increased availability of $NH_4^+$-N, nitrification into $NO_3^-$-N was unaffected with incubation.

The increase in mineral N availability, at least from the PL, would be in line with the findings by Chan et al. [57], who concluded that greater radish yields following poultry litter-based biochar applications were due to enhanced N availability in acidic, nutrient-poor Alfisol. In contrast to the current findings, studies with wood-derived biochars or recently formed charcoals created by wildfire indicated increased nitrification rates in forest soils and shifts in community composition towards microbial nitrifiers [58,59]. More specifically, the activity of soil nitrifiers was similarly increased in sandy loam soils after receiving up to 5 Mg ha$^{-1}$ of a poultry litter biochar; however, in contrast to the current study, the soil remained acidic (pH < 6.6) [46]. Potentially, $NH_3$ emissions might have prevented nitrification, at least in treatments with the highest PL application rates. However, the significantly elevated C:N ratio in both PC and PL treatments compared to NC0 controls might also explain why this did not happen. In a greenhouse study, high C:N was blamed for the significantly reduced N availability following the addition of a wood-derived charcoal (10% w/w) to both a clay forest Anthrosol and sandy agricultural Ferrasol [60]. Alternatively, ammonification or nitrification processes might have been inhibited by sorption of mineral N to biochar surfaces [61–63]. Additionally, the soil nitrifier community might not have tolerated the increased alkalinity, particularly with PL biochar addition to the Tifton soil [64].

### 4.3. Priming Effects

The potential that adding PC and PL biochars would induce priming effects, which result in turnover of native soil organic matter or biochar into mineralized or processed forms of C and N, was investigated. Individually, loss of soil C or increases in soluble OC, microbial biomass, and soil respiration have been used to indicate C priming [19–21]. In the current study, changes in all but soil respiration, either initially or following incubation, provided some indication of C priming, particularly SOC loss. Only a few indicators were consistent with respect to treatments. For example, SOC loss occurred in PC treatments but increases in both soluble OC and MBC occurred in initial Tifton PL treatments. This apparent C priming did not continue as soluble OC and MBC did not increase in most treatments following the 28-day incubation. The inability to capture an increase in C mineralization to $CO_2$-C, despite a loss of SOC, with weekly respiration measurements also suggested a rapid but short-term priming effect. This short-term priming contrasts with other studies, where short-term priming of C mineralization lasted up to 42 days with either fresh and 4-month lab-aged biochar [65] or up to 90 days with low temperature biochars (<400 °C) [66]. The N priming pathway was evident as an immediate increase in $NH_4^+$-N in initial Cecil PL treatments and the associated increase in $NO_3^-$-N (thought it did not change with incubation) in Cecil PL and Tifton PC and PL treatments. More importantly, this form of N priming continued in Tifton PL treatments through the 28-day incubation, because ammonification was evident as increased $NH_4^+$-N in the final sample analysis. This ammonification activity was probably related to increased MBC and availability of soluble OC.

Determining if C or N priming was acting on native organic matter or the added biochar was partly definitive. In the Tifton PL200 treatment, the negative change in soluble OC after incubation (95 mg kg$^{-1}$ soil) was twice the initial content in the Tifton NC0 control but only 36% of the initial soluble OC content, suggesting the biochar was the C source. In contrast, total SOC loss in Cecil PC treatments with no apparent change in soluble OC, MBC, or basal respiration makes it difficult to determine the C source. In contrast, N priming appeared to be acting on the added biochar primarily due to the significant addition of $NH_4^+$-N from PL and apparent immobilization of N due to increased C:N ratios from PC. Determining the priming action on native organic matter or added biochar would

be best accomplished by using stable isotope tracers, which have been used to indicate losses from soil organic matter, added biochar, or both [66–68].

*4.4. Utility of Short-Term Incubations*

An evaluation of the literature indicates many gaps in understanding and predicting outcomes from biochar use in the environment [6]. Most can be linked to the variability in feedstocks, pyrolysis conditions, soil types, and climates [1,34]. As the above discussion indicates, only two changes measured here were consistent with the use of biochars as a soil amendment: a change in soil pH (i.e., liming effect [20,69]) and an increased soil organic C content [1,2]. Although the short-term incubation does not provide a sufficient basis to predict a long-term response with field deployment, the results provide important outcomes to consider before making field application recommendations:

1.  Soil pH changes. Application of the generally neutral pH PC biochar had a moderate effect on the moderately acidic Cecil soil and highly acidic Tifton soil, regardless of the application rate. Conversely, the highly alkaline PL biochar had significant effects on both soils in generating alkaline biochemical behaviors (e.g., $NH_3$ production [54]) at increasing application rates. As soil pH did not decline during this 28-day incubation, the application rates of PL biochar should probably be limited, unless transience is determined by further investigation.

2.  Nutrient availability. The current incubation study supplied sufficient information to determine if the biochar tested would improve N availability. Additionally, the incubation indicated that this aspect required interpretation with respect to changes in soil pH. Of particular concern was the potential for $NH_3$ volatilization, which was probably the main reason nitrification processes were inhibited. Of important note, biochar pH can be controlled by pyrolysis temperature and feedstock selection [6]. In the Mandal et al. [56] study cited above, better reduction of $NH_3$ volatilization was provided by the poultry manure biochars produced at or below 350 °C compared to those produced at or above 450 °C due to a greater increase in soil pH from these biochars.

3.  C sequestration. Both biochars initially increased soil organic C, but only the PL biochar added a soluble source of OC that was probably available to the microbial community. The microbial community was generally unaffected by the addition of PC biochar to either soil. With lack of substantial response in N processes, and the short-term nature of the C priming effect probably responsible for the SOC loss, this biochar could be considered generally inert to microbially driven processes, making it a more appropriate amendment to improve the C sequestration capacity of either soil. In contrast, the PL biochar did add soluble OC, which resulted in an initial increase in microbial biomass, but due to the observed lack of impact on nitrification processes, it was probably negatively affected diversity-wise by further incubation under adverse conditions. Due to the complexity of C and N processes exposed with application of the PL to the Tifton soil, additional lab or small-scale field incubations are probably necessary to determine its true nature before large-scale deployment was conducted.

In summary, the small-scale incubations used in this study indicated that PC biochar would probably have positive effects if added to either of these soil types at the tested application rates. On the other hand, caution is warranted with PL biochar use. For the less acidic Cecil soil, negative outcomes were limited, probably due to the less drastic change in soil pH. However, in Tifton soil, though PL biochar was a good provider of potentially available nutrients, the smaller initial microbial community in this more degraded soil might be more compromised by its addition.

## 5. Conclusions

The results presented here indicated that C and N dynamics differed among the soil types and biochar treatments. Initial impacts were generally predictable based on the characteristics of soils and biochars prior to mixing; for example, the alkaline PL biochar

significantly increased the pH in the acidic Tifton soil, and both biochars acted like typical organic soil amendments that increase soil C content. In Cecil soils, C and N dynamics were somewhat similar following application of either of the two biochars. In contrast, in Tifton soils, C and N dynamics were different between the two biochar treatments. From the biochar perspective, application of PC biochar induced changes in C and N dynamics only when applied to Cecil soil and not Tifton soil. In contrast, application of the PL biochar generally induced the opposite effect on C and N dynamics between the two soil types.

Impact on N dynamics were more prominent than impacts on C dynamics, as $NH_4^+$-N and $NO_3^-$-N-related processes were influenced either initially or after incubation in all biochar treatments. With PC biochar, C and N were probably sourced from the soil, but with PL biochar, C and N were probably sourced from the biochar. As C priming was short-term, the addition of PC, particularly in Tifton soil, would probably be suitable for long-term C sequestration outcomes. The addition of either PC or PL biochar would probably be suitable for enhancing N availability, but application rates need to be considered given the potential for increased alkalinity and ammonia volatilization when applied to acidic soils such as Tifton.

**Author Contributions:** Conceptualization, S.L.W. and J.W.G.; methodology, S.L.W., K.C.D. and J.W.G.; formal analysis, S.L.W.; investigation, A.M.L.; writing—original draft preparation, S.L.W.; writing—review and editing, S.L.W., K.C.D., J.W.G. and A.M.L. All authors have read and agreed to the published version of the manuscript.

**Funding:** The authors acknowledge the financial support of the U.S. Department of Defense, Centers for Research Excellence in Science and Technology ARO Grant, W911NF-11-1-0218.

**Data Availability Statement:** The data presented in this study are available on request from the corresponding author.

**Acknowledgments:** We greatly appreciate the assistance of Alan Wilts, Nancy Barbour, and Tyson Mastin at the USDA ARS Morris, MN, and Keith Harris and Brian Bibens at the University of Georgia for their contributions in completing this experiment. The USDA is an equal opportunity provider and employer.

**Conflicts of Interest:** The authors declare no conflict of interest.

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
