# Peer review of "Pine Chip and Poultry Litter Derived Biochars Affect C and N Dynamics in Two Georgia, USA, Ultisols"

_agronomy, doi:10.3390/agronomy13020531_

Round 1
Reviewer 1 Report
The authors presented a manuscript on C and N dynamics in two Ultisols under Pine chip and poultry litter-derived biochars. The study is not novel because the impact of biochar and C and N dynamics have been widely studied. However, the study can provide incremental knowledge on the impacts of different biochars on the same soil type under similar conditions. However, I have the following concerns the authors need to address to make it publishable.
First, the in-text citation is not consistent, in some cases, numbers as in line 33, and in other cases, as shown in the other parts of the manuscript. Besides, the format used in the manuscript is not the format used in “Agronomy”. A thorough revision needs to be done
The CEC of biochar, surface area, porosity, and functional groups on the surface has a significant impact on its N retention potentials. These properties should be provided to give more credence to the observations and to support the observations.
The analyzed data showing the difference in means, SE, and means should be preferably presented using figures. Figures can make the results easily understandable. Besides, the presentation of the results makes its visualization rather difficult.
The use of percentage in biochar application rate looks misleading. It is best to present the percentage of biochar in relation to the soil or the rate of biochar applied, and not how it was presented (0-200%)
The conclusion is presented poorly. The author was repeating the results and discussion in this section, which is not proper. It must be revised and presented briefly. Additional information on the conclusion can be found in my comments below.
Additional comments
Line 10: write C in full at the first mention
Line 14: write N in full at the first mention
Line 14-15: write the formula for ammonium and nitrate properly
Line 133: how was the CO2 calculated? What formula was used and its reference?
Table 1: write the formula for ammonium and nitrate properly
The units should have an underline before the values in the tables. The same applies to other tables
The mean values should have mean separation using lowercase alphabets. It will highlight the significant differences.
The biochar rates presented in Table 1 should be in the proportion of soil. 200% is not an ideal presentable. Besides the g/kg used on this table is inappropriate based on the initial presentation on line 100%
Line 226: write carbon dioxide correctly
Line 226: Does Table 3 show the cumulative CO2 respiration? I doubt. Besides, there should be a figure showing the CO2 emission trend over the sampling period and then, the cumulative value.
Line 231-233: How is it that there is no change in total N after biochar application? Biochar contains a considerable amount of N which has often increased the total N status of soils. Besides the manner of the presentation of the results makes it rather difficult to visualize your results. The use of figures will be more appropriate.
Line 234: write correctly as NO3- -N. Do the same to NH4+-N and correct this error across the manuscript.
Line 245: write the formula for ammonium and nitrate properly
Line 297: was the biochar applied as a percentage or in g/kg as shown in the tables? Please revise carefully across the manuscript.
The conclusion is unreasonably long. The conclusion should only highlight what was observed, and the mechanisms regulating it. Hence, it should be reasonably summarized.
Most of the explanations in the conclusion can be incorporated into the discussion.
Author Response
Response to reviewer 1
The authors presented a manuscript on C and N dynamics in two Ultisols under Pine chip and poultry litter-derived biochars. The study is not novel because the impact of biochar and C and N dynamics have been widely studied. However, the study can provide incremental knowledge on the impacts of different biochars on the same soil type under similar conditions. However, I have the following concerns the authors need to address to make it publishable.
First, the in-text citation is not consistent, in some cases, numbers as in line 33, and in other cases, as shown in the other parts of the manuscript. Besides, the format used in the manuscript is not the format used in “Agronomy”. A thorough revision needs to be done.
As the journal allows for free formatting of the initial submission, we did not use the number formatting. We have since reformatted the references in the revision.
The CEC of biochar, surface area, porosity, and functional groups on the surface has a significant impact on its N retention potentials. These properties should be provided to give more credence to the observations and to support the observations.
We have added the electrical conductivity measurements, and estimated CEC. We agree that all these properties can have significant impacts, but we do not have all this information. We no longer have this biochar to send for additionally analyses.
The analyzed data showing the difference in means, SE, and means should be preferably presented using figures. Figures can make the results easily understandable. Besides, the presentation of the results makes its visualization rather difficult.
At the request of the academic editor, we revised all the results to show means. As we have so many soil properties, due to the complexity of making comparisons to show differences among means figures were not used as these differences were difficult to properly show on a figure.
The use of percentage in biochar application rate looks misleading. It is best to present the percentage of biochar in relation to the soil or the rate of biochar applied, and not how it was presented (0-200%)
We understand that this could be misleading, but we disagree with revising this presentation. For one – the biochar to soil ratio differs because the two soils have drastically different bulk densities. As we applied the biochar at a consistent application rate in grams based on area coverage (hectares) other ways to present the information could also be construed. We have revised the abstract to reflect that application rates reached 90 kg per hectare, and we provide a clear indication of the total grams of biochar and their ratio to soil in the methods. We did not use the percentage designation in the abstract to avoid misleading casual readers.
The conclusion is presented poorly. The author was repeating the results and discussion in this section, which is not proper. It must be revised and presented briefly. Additional information on the conclusion can be found in my comments below.
The conclusion was drastically reduced in the second revision and more streamlined to address only the goals rather than to repeat results or discussion.
Additional comments
Line 10: write C in full at the first mention
Line 14: write N in full at the first mention
Done for both C and N, we have also written out fully all other nutrients mentioned in the manuscript.
Line 14-15: write the formula for ammonium and nitrate properly
We have changed all instances to reflect the valence characteristics of these N forms.
Line 133: how was the CO2 calculated? What formula was used and its reference?
Revised to: Total soil respiration in total g CO2-C per kg soil emitted over 28-days was calculated for each replicate chamber by summing the estimated average daily rate for each week assuming a linear change from week to week. This was done by calculating daily respiration at the midpoint of the slope (ca. day 4) from day 0 to day 7 of the week. This is equivalent to calculating the area under the curve generated by the graph of respiration over time.
We did not cite a reference as this is a standard mathematical approach to determine the area under the curve. As these were straightforward linear changes the slope of the line between weeks was calculated from y=mx+b and the midpoint determined by x.
Table 1: write the formula for ammonium and nitrate properly
Done for all tables and in text revisions
The units should have an underline before the values in the tables. The same applies to other tables
We accept the suggestion
The mean values should have mean separation using lowercase alphabets. It will highlight the significant differences.
In the revision mean separations for significant two-way interactions (Tables 2 and 3) were done with lower case letters. In Table 4, the three-way interaction is complex (letters a through k) and would crowd the table. We took the approach of evaluating the main effects within the three-way and express sampling time differences with a *, soil differences with a # and treatment differences (as in the other tables) with letters.
The biochar rates presented in Table 1 should be in the proportion of soil. 200% is not an ideal presentable. Besides the g/kg used on this table is inappropriate based on the initial presentation on line 100%
We understand your point of view and have responded to this same suggestion above. We now just use a “treatment” designation.
Line 226: write carbon dioxide correctly
Done
Line 226: Does Table 3 show the cumulative CO2 respiration? I doubt. Besides, there should be a figure showing the CO2 emission trend over the sampling period and then, the cumulative value.
The revised table presents the total amount of CO2-C calculated to have been emitted over the entire 28-day incubation. After analysis the soil*treatment interaction was significant, but multiple comparisons indicate few meaningful differences, in that no patterns for soil, application rate or biochar stood out. Due to this, displaying the data in other forms seemed irrelevant.
Line 231-233: How is it that there is no change in total N after biochar application? Biochar contains a considerable amount of N which has often increased the total N status of soils. Besides the manner of the presentation of the results makes it rather difficult to visualize your results. The use of figures will be more appropriate.
As explained above, using figures to display the results for all soil properties would be overwhelming for the manuscript as a whole, at least nine figures if all were limited to the three-way interaction. We prefer the more condensed approach of using tables.
Line 234: write correctly as NO3- -N. Do the same to NH4+-N and correct this error across the manuscript.
Corrected throughout
Line 245: write the formula for ammonium and nitrate properly
Done
Line 297: was the biochar applied as a percentage or in g/kg as shown in the tables? Please revise carefully across the manuscript.
We apologize for the confusion. In our experience units are displayed under the line and apply only to the columns that the units hover over. We now provide units in a different way to avoid this confusion.
The conclusion is unreasonably long. The conclusion should only highlight what was observed, and the mechanisms regulating it. Hence, it should be reasonably summarized.
Most of the explanations in the conclusion can be incorporated into the discussion.
We have condensed both the discussion and conclusions sections
Reviewer 2 Report
The manuscript entitled ''Pine chip and poultry litter derived biochars affect C and N dynamics in two Georgia, USA, Ultisols'' investigated the responsibility in C and N dynamics relied on soil type, biochar type, or application rate and assess possible biochar priming effects on appropriateness for field application through delivery of nutrients or carbon sequestration qualities. The main problem of the manuscript is numerous writing and structural mistakes.
Professional editing is recommended. Please revise all the manuscript for grammatical mistakes and English polishing also the paper contains some long and repetitive sentences.
Please see the comments below:
Abstract:
The abstract has problems in terms of writing and most sentences are not understandable and should be rewritten.
Lines 11-12: This is a general statement and not true for all biochars. Please correct the sentence.
Line 12: “we used” is not scientific à you can say “in this research…” and rewrite the sentence.
Introduction:
Lines 46-47: provide source(s)
Lines 50-52: There are problem with the structure of the sentence and should be rewrite.
Line 66: “we used” is not scientific à you can say “in this research…” and rewrite the sentence.
Please check it in full text.
Introduction is not enough and needs more extend. At the end of this section the novelty of study is not clear. Research background is poor. Authors should better explain the novelty of the study and what it offers in comparison to previous literature. Here is a newly published work that fit with your scope. You can use it: https://doi.org/10.3390/su142214722
Materials and Methods:
It is better to divide this section into different parts and explain the method for each measurement separately and references should be given for the methods used.
Please add model and type of the devices used for your measurements.
Lines 79-84: provide source(s)
Line 88: 0.5 hà it’s a very short time for complete pyrolysis. Please check it and add reference(s) for this sentence.
Line 134: “was”àwere
Results and Discussion
Results and Discussion are well described.
There is only one point: “References are not up to date and should be updated”.
References
The reference style is NOT correct. please check it based on the instructions for the Authors.
Author Response
Response to reviewer 2.
The manuscript entitled ''Pine chip and poultry litter derived biochars affect C and N dynamics in two Georgia, USA, Ultisols'' investigated the responsibility in C and N dynamics relied on soil type, biochar type, or application rate and assess possible biochar priming effects on appropriateness for field application through delivery of nutrients or carbon sequestration qualities. The main problem of the manuscript is numerous writing and structural mistakes.
Professional editing is recommended. Please revise all the manuscript for grammatical mistakes and English polishing also the paper contains some long and repetitive sentences.
The manuscript has undergone extensive editor to correct any grammatically mistakes, shorten sentences and reduce repetition.
Please see the comments below:
Abstract:
The abstract has problems in terms of writing and most sentences are not understandable and should be rewritten.
Lines 11-12: This is a general statement and not true for all biochars. Please correct the sentence.
Corrected to: “Some biochars…have the potential…”
Line 12: “we used” is not scientific à you can say “in this research…” and rewrite the sentence.
Corrected to “A 28-day lab incubation was used to assess…”
Introduction:
Lines 46-47: provide source(s)
Added citation.
Lines 50-52: There are problem with the structure of the sentence and should be rewrite.
Original: Studies indicated that microbial stimulation is due to a C priming effect, whereby addition of biochar increased microbial turnover, measured as mineralization of C into CO2, of either the added biochar or native soil organic matter
Revised: Studies have defined C priming effect as a stimulation of microbial turnover of C into CO2, where the C is sourced from the biochar or from native soil organic matter.
Line 66: “we used” is not scientific à you can say “in this research…” and rewrite the sentence.
Revised to: “Mesocosm incubations were used to…”
Please check it in full text.
We corrected all in-text first person instaces.
Introduction is not enough and needs more extend. At the end of this section the novelty of study is not clear. Research background is poor. Authors should better explain the novelty of the study and what it offers in comparison to previous literature. Here is a newly published work that fit with your scope. You can use it: https://doi.org/10.3390/su142214722
We have added several more citations in the introduction and discussion, and tried to find those that were published since 2019.
Materials and Methods:
It is better to divide this section into different parts and explain the method for each measurement separately and references should be given for the methods used.
Please add model and type of the devices used for your measurements.
We divided the section into experimental set up and “sampling and analyses”. Sampling and analyses we separate each approach into a new paragraph and added more complete information on analytical equipment models and methods including citations where needed.
Lines 79-84: provide source(s)
We added information on the soil sources.
Line 88: 0.5 hà it’s a very short time for complete pyrolysis. Please check it and add reference(s) for this sentence.
These were small batches produced at bench scale the holding time was sufficient for pyrolysis. A relevant citation was added.
Line 134: “was”àwere
We revised the paragraph to improve clarity of the statistical approach.
Results and Discussion
Results and Discussion are well described.
Thank you for this comment
There is only one point: “References are not up to date and should be updated”.
We agree and have added some updated references.
References
The reference style is NOT correct. please check it based on the instructions for the Authors.
The journal allows free-formatting for the initial submission. We have now corrected the Reference style with numbers according to the journal’s Author instructions https://www.mdpi.com/journal/agronomy/instructions was not required for initial submission.
Round 2
Reviewer 1 Report
The manuscript has be thoroughly revised according to the review comments.
I have a few minor comments
Table 4: The additional explanation aside the title of the table should preferable at the bottom of the table as a footnote as done in 3.
Reviewer 2 Report
The manuscript has been well edited and there are no more comments.